# Acute Myeloid Leukemia Causes Serious and Partially Irreversible Changes in Secretomes of Bone Marrow Multipotent Mesenchymal Stromal Cells

**DOI:** 10.3390/ijms24108953

**Published:** 2023-05-18

**Authors:** Aleksandra Sadovskaya, Nataliya Petinati, Nina Drize, Igor Smirnov, Olga Pobeguts, Georgiy Arapidi, Maria Lagarkova, Alexander Belyavsky, Anastasia Vasilieva, Olga Aleshina, Elena Parovichnikova

**Affiliations:** 1National Medical Research Center for Hematology, 125167 Moscow, Russia; 2Department of Immunology, Faculty of Biology, Federal State Budget Educational Institution of Higher Education M.V. Lomonosov Moscow State University, 119991 Moscow, Russia; 3Lopukhin Federal Research and Clinical Center of Physical-Chemical Medicine of Federal Medical Biological Agency, 119435 Moscow, Russia; 4Shemyakin-Ovchinnikov Institute of Bioorganic Chemistry of the Russian Academy of Sciences, Miklukho-Maklaya 16/10, 117997 Moscow, Russia; 5Engelhardt Institute of Molecular Biology, Russian Academy of Sciences, 119991 Moscow, Russia

**Keywords:** multipotent mesenchymal stromal cells, secretome, acute myeloid leukemia

## Abstract

In patients with acute myeloid leukemia (AML), malignant cells modify the properties of multipotent mesenchymal stromal cells (MSCs), reducing their ability to maintain normal hematopoiesis. The aim of this work was to elucidate the role of MSCs in supporting leukemia cells and the restoration of normal hematopoiesis by analyzing ex vivo MSC secretomes at the onset of AML and in remission. The study included MSCs obtained from the bone marrow of 13 AML patients and 21 healthy donors. The analysis of proteins contained in the MSCs-conditioned medium demonstrated that secretomes of patient MSCs differed little between the onset of AML and remission; pronounced differences were observed between MSC secretomes of AML patients and healthy donors. The onset of AML was accompanied by a decrease in the secretion of proteins related to ossification, transport, and immune response. In remission, but not at the onset, secretion of proteins responsible for cell adhesion, immune response, and complement was reduced compared to donors. We conclude that AML causes crucial and, to a large extent, irreversible changes in the secretome of bone marrow MSCs ex vivo. In remission, functions of MSCs remain impaired despite the absence of tumor cells and the formation of benign hematopoietic cells.

## 1. Introduction

The bone marrow microenvironment plays a central role in maintaining the dynamic balance between hematopoietic stem and progenitor cells (HSPCs) self-renewal, differentiation, quiescence, and proliferation in homeostatic and pathologic conditions [1]. This microenvironment is composed of multiple cell types, the most prominent of which are considered to be specific subsets of multipotent mesenchymal stromal cells (MSCs) or derived from them [2]. Bona fide MSCs are rare elements in the bone marrow (0.01%) [3], which are able to differentiate into all types of stromal cells (fibroblasts, osteoblasts, adipocytes, etc.). Within the bone marrow, MSCs are prominently involved in orchestrating the behavior of hematopoietic stem and progenitor cells, ensuring a lifelong blood supply [2,4]. The MSCs-related cells, in particular the CAR cells (CXCL12-abundant reticular cells), the nestin+ cells, and the CD146+ cells, play a major role in the communication between the bone marrow HSPCs and the microenvironment while displaying significant overlap with each other [5]. MSCs secrete cytokines, chemokines, and growth factors, as well as exosomes and microvesicles containing proteins and genetic material, collectively referred to as the MSC secretome [6,7]. These factors signal to bone marrow cells in a manner that can increase proliferation [8], direct migration [9], initiate differentiation [10], modulate activation or polarization of immune cells, resolve inflammation, deposit matrix, and heal wounds [11]. Based on these findings, the ISCT updated in 2019 the definition of MSCs to include functional assays, such as the trophic factor secretion, modulation of immune cells, and promotion of angiogenesis [12]. A number of cytokines, cytokine receptors, and adhesion molecules have been implicated in the cross-talk between HSPCs and cells of the microenvironment, involving in particular signaling axes CXCL12 (SDF-1)—CXCR4, SCF (stem cell factor)—c-kit, vascular cell adherence molecule 1 (VCAM1)—VLA-4 (α4β2), angiopoietin-1 (Ang-1) and Tie-2, thrombopoietin (TPO)—MPL [13]. In vitro studies showed that MSCs and osteoblasts secrete hematopoietic cytokines such as CXCL12, SCF, Ang-1 IL-6, and express Notch ligand Jagged 1 [14,15]. SCF and IL-6 support normal hematopoiesis but also the maintenance and quiescence of HSCs. SCF is primarily expressed by the perivascular cells in the bone marrow [16]. Ang-1 regulates HSCs quiescence and has an anti-apoptotic effect [13]. TGFβ1 (transforming growth factor β1) has also been demonstrated to induce HSCs quiescence [17]. Osteopontin (OPN) is a glycoprotein synthesized by osteoblasts in the endosteal region that recognizes the integrin αvβ1 on the HSC surface and plays a role in cell adhesion, inflammatory responses, and angiogenesis. The secretome of MSCs may reflect changes in the regulatory functions of these cells caused by the malignant transformation of hematopoietic cells.

Acute myeloid leukemia (AML) is a malignant disease characterized by the block of HSPC differentiation and active proliferation of clonal malignant HSPCs, which rapidly leads to bone marrow failure and eventually to death if left untreated. During leukemogenesis, AML cells (or blasts) progressively occupy and likely alter the bone niche where normal HSCs reside [18]. Emerging evidence suggests that leukemia cells induce molecular changes in distinct hematopoietic and non-hematopoietic cell populations in the niche. These changes contribute to the transformation of the normal hematopoietic niche into a ‘leukemia niche’ that becomes permissive to leukemia growth and disrupts normal hematopoiesis [19]. Disruption of the receptor-adhesion molecule interactions between HSC niches and leukemic stem cells might be a therapeutic target, as discussed [20,21]. MSCs isolated from AML patients generate various factors that stimulate AML cell homing into hematopoietic stem cell niches in bone marrow, resulting in AML cell survival [22]. AML-MSCs are transcriptionally [23], genetically [24,25], and functionally [26] distinct from healthy donor counterparts. MSCs from AML patients downregulate hematopoietic maintenance and homing factors and show adipogenic and osteogenic differentiation [27] deficiencies that change the composition of the cellular niche [23].

Several studies suggest that extrinsic, microenvironmental changes in the bone marrow may also promote the transition from malignant precursor cells to active disease [28,29]. However, the precise mechanisms through which leukemia cells co-opt and modify the normal hematopoietic niche remain largely unknown. When remission is achieved, it might be expected that in the absence of tumor cells, the stromal microenvironment will recover to its pre-disease state. However, one must also keep in mind that changes in MSCs caused by leukemia may cooperate with factors associated with chemotherapeutic treatment.

The aim of this study was to elucidate changes in the MSCs secretome occurring in patients with AML and during the remission of the disease. Our data demonstrate that the MSC secretome in remission, although it differs from the one at the onset of the disease, does not show significant recovery and still differs substantially from the MSC secretome in healthy donors.

## 2. Results

MSCs were obtained from bone marrow mononuclear cells collected from patients at diagnosis (referred to as AML-MSCs), at the time of remission (R-MSCs), and from age-matched healthy donors (D-MSCs). A summary of the data is presented in Figure 1.

### 2.1. Comparison of AML-MSC and D-MSC Secretomes

The work analyzed 2833 proteins identified in the secretomes of the studied MSCs. Among these, 685 proteins significantly differed in AML-MSCs, R-MSCs, and D-MSCs. Secretion levels for 533 protein were different in AML compared to donors but not in remission (Appendix A).

AML-MSCs lost secretion of five proteins compared to D-MSCs: these are proteins associated with the extracellular region (NCAM1, F10, CST6, A1BG, LOXL4).

When comparing upregulated genes in AML-MSCs versus D-MSCs—AML-MSCs secreted 141 proteins significantly more actively than D-MSCs. Among them are proteins that are important for maintaining hematopoiesis and the immune response, forming relationships—HLA-A, MIF, FN1, COL8A1, POSTN, IGFBP5, CD44, LGALS3 и ANXA1, CXCL12 (Appendix A).

When comparing down-regulated genes in AML-MSCs versus D-MSCs—AML-MSCs secreted 343 identified proteins significantly lower than D-MSCs—COL6A1, TGFB, GAS6, PDGFA, PDGFRB, SERPINA3, SERPINA7, SERPINE2, VCAM1, CFH, C3, and others (Appendix A).

Principal component analysis (PCA) indicates significant differences in the secretome of AML-MSCs and D-MSCs. Principal components 1 to 3 were plotted against each other (Appendix A). Partial Least Squares Discriminant Analysis (PLS–DA), based on non-normalized log_10_ intensities, revealed significant changes in component 1 versus 2 and lower pronounced alterations when compared to component 1 versus 3 and component 2 versus 3 (Appendix A). The most significant differences in protein secretion are shown in Volcano plots for AML-MSCs versus D-MSCs (Figure 2A).

GO-term enrichment analysis for compartment, biological process, and cellular component of AML-MSCs versus D-MSCs is presented in Table 1. Many different proteins involved in various spheres of cell activity are changed in patients’ MSCs. Proteins involved simultaneously in various processes change their expression in AML-MSCs (Figure 2B,C). The changes are associated with the extracellular space, matrix, exosomes, and vesicles. Many changes are associated with the involvement of MSCs in the maintenance of hematopoietic cells and the immune response.

### 2.2. Comparison of R-MSCs and D-MSCs Secretome

Comparison of the secretomes of R-MSCs and D-MSCs revealed significant differences in 550 proteins. Obviously, there was no recovery of secreted proteins when remission was achieved. There was no recovery of secretion of proteins whose expression was lost in AML-MSCs (NCAM1, F10, CST6, A1BG, LOXL4). R-MSCs began to secrete certain proteins not secreted in D-MSCs: 61 proteins (21 genes mapped), including CTGF, SERPINE1, MMP11, PTKSB, and CSF1.

Principal component analysis (PCA) indicates significant differences in the secretome of R-MSCs and D-MSCs. Principal components 1 to 3 were plotted against each other (Appendix A). Partial Least Squares Discriminant Analysis (PLS–DA), based on non-normalized log_10_ intensities, revealed significant changes similar to observed when compared R-MSCs and D-MSCs (Appendix A). The most significant differences in protein secretion are shown in Volcano plots for R-MSCs versus D-MSCs (Figure 3A).

Simultaneously, proteins involved in compartments and various processes change their expression in R-MSCs in comparison with D-MSCs (Figure 3B,C). The changes are associated with the extracellular space, matrix, exosomes, and vesicles. Many changes are associated with cell adhesion, differentiation, aging, cytokine-mediated response, and the immune system.

GO-term enrichment analysis for the biological process of R-MSCs versus D-MSCs is presented in Table 2.

Thus, upon reaching remission, the composition of the R-MSCs secretome does not recover, and it continues to differ from the secretome of D-MSCs.

### 2.3. Comparison of AML-MSCs and R-MSCs Secretome

Comparison of the secretomes of AML-MSCs and R-MSCs revealed similar and different proteins in secretion (Figure 4 and Appendix A).

GO-term enrichment analysis for biological process of AML-MSCs versus R-MSCs presented in Table 3.

Expressed both in AML-MSCs and R-MSCs and not expressed in D-MSCs18 proteins (17 mapped genes ANKRD20A8P, B2M, BCAT1, C1S, HLA-A, MIF, CHIC2, CHMP1A, HAGH, PPARGC1A, PPP1R3F, PTK2B, SRCAP, SYT7, and TCTN3).

Secreted only by AML-MSCs 26 proteins (21 mapped genes), only in R-MSCs 43 (26 mapped genes), and not in D-MSCs.

Secreted in AML-MSCs, not secreted in R-MSCs: 255 proteins, significant only 3 proteins—UCL1—Mucin-like protein 1; May play a role as a marker for the diagnosis of metastatic breast cancer; SSBP1—Single-stranded DNA-binding protein, mitochondrial; this protein binds preferentially and cooperatively to ss-DNA. Probably involved in mitochondrial D; UBE2I—SUMO-conjugating enzyme UBC9; Accepts the ubiquitin-like proteins SUMO1, SUMO2, SUMO3, and SUMO4 from the UBLE1A-UBLE1B E1 complex.

Secreted in R-MSCs, not secreted in AML-MSCs: 405 proteins, significant only 3: F10—Coagulation factor X; Factor Xa is a vitamin K-dependent glycoprotein that converts prothrombin to thrombin in the presence of factor Va; CTGF—Cellular communication network factor 2; Connective tissue growth factor; Major connective tissue mitoattractant secreted by vascular endothelial cells; LAMA2—Laminin subunit alpha-2; Binding to cells via a high-affinity receptor, laminin is thought to mediate the attachment, migration, and organization of cells.

The number of proteins with different secretion in AML-MSCs and R-MSCs compared to D-MSCs and their main functions is presented in Figure 1, Table 4, Appendix A.

Significantly lower in AML-MSC versus R-MSCs 13 proteins (P4HB, SOD1, HEL-S-44, SDF4, PDIA3, HEL-S-269, PAWR, SGSH, LMNA, S100A13, CTSD, HEL-S-130P, TNFRSF11B, AXL, PTGDS, and RCN1).

Significantly higher in AML-MSC versus R-MSCs 21 proteins (EXT2, ADAMTS2, LOX, LTBP3, RPL7A, NID2, PRKCSH, ARHGDIA, HEL-S-47e, TUBB6, CPQ, PGCP, PGM2, CFAP44, ERP29,HEL-S-107, PSMD6, LGALS1, VTN, PSMB7, PDIA3, CTSL, INHBA, and MFAP5).

Among 86 proteins higher expressed in AML-MSCs and R-MSCs than in D-MSCs, the expression of 40 proteins decreased in remission: FTO, CAVIN3, CTSV, LMAN1L, CLTB, GAPD, HNRNPA3, IGFBP5, GSN, PPIA, COL18A1, PODN, CXCL12, LSM1, HLA-DRB1, HMGN1, NME1, PSMA1, PSMB5, PDIA3, PCLO, SEC23, PRSS3, GDI1, RTN4, SSBP1, SLC30A9, TSSK4, TPM2, LSM3, UBE2V1, ISG15, and others. The expression of 18 proteins increased even more: AKR1A1, CHST15, CCL2, COL8A1, FN1, POSTIN, HMGN2, PSPH, YIPF3, TK2, TAGLN3 and others (Appendix A).

Among 220 proteins lower expressed in AML-MSCs and R-MSCs than in D-MSCs, only one protein decreased even more in remission-ERP29, 8 proteins significantly increased in remission but did not reach normal levels—ADAMTS2, INHBA, EXT2, LTBP3, LOX, PGM2, TUBB6, AXL. The secretion of 125 proteins did not change or decrease even more. The secretion of other proteins increased insignificantly in R-MSCs but did not reach the level of donors.

GO-term enrichment analysis for the biological process of AML-MSCs and R-MSCs versus D-MSCs is presented in Table 4.

## 3. Discussion

The tumor microenvironment is considered nowadays as one of the main players in cancer development and progression. AML microenvironment is highly complex and includes MSCs, their progeny, and a large list of extracellular matrix proteins and soluble factors secreted by AML-MSCs.

Comparative studies of the MSC secretome revealed large differences between the secretomes of AML-MSCs and D-MSCs, as well as the absence of the full restoration of the MSC secretome profile after treatment and upon reaching remission. D-MSCs did not secrete PPARGC1A -Peroxisome proliferator-activated receptor gamma coactivator 1-alpha. This protein is involved in the immune response and is a pivotal transcriptional coactivator regulating mitochondrial biogenesis and metabolism [30]. PPARGC1A greatly increases the transcriptional activity of PPARG and thyroid hormone receptors.

As compared to D-MSCs, the secretion of six proteins was not revealed in AML-MSCs. One of them is NCAM1—Neural cell adhesion molecule 1 (CD56), representing a transmembrane glycoprotein modulating cell–cell and cell–matrix interactions. NCAM1 expression is strongly associated with constitutive activation of the MAPK-signaling pathway, regulation of apoptosis, or glycolysis [31]. Bone marrow-derived MSCs from NCAM-deficient mice exhibit the defective migratory ability and significantly impaired adipogenic and osteogenic differentiation potential. The mechanism governing NCAM1-mediated migration of MSCs involves the cross-talk between NCAM1 and fibroblast growth factor receptor (FGFR), which in turn activates MAPK/ERK signaling and, thereby, the migration of MSCs [32]. Clinical-grade human bone marrow MSCs were previously shown to express NCAM1 isoforms on their surface [33]. The presence of NCAM1 in a secreted fraction is likely to reflect either its inclusion in microvesicles produced by MSCs [34] or shedding by extracellular proteases [35]. Thus, the complete absence of secretion of this protein in AML-MSCs may indicate strongly reduced production of this protein and, thus major changes in cell–matrix interactions in these cells.

Another protein with strongly reduced secretion is CST6—Cystatin-E/M belongs to the type 2 cystatin family, which are mainly extracellular polypeptide inhibitors of cysteine proteases acting to prevent excessive proteolysis. Three proteases, including cathepsin B (CTSB), cathepsin L (CTSL), and legumain (LGMN), are known to be inhibited by CST6 in human cells. Several peptides mimicking the function of CST6 are able to suppress cancer cell-induced osteoclastogenesis and bone metastasis [36]. These findings reveal the CST6-CTSB signaling axis in osteoclast differentiation and provide a promising approach to treating bone diseases with CST6-based peptides [36]. Both recombinant CST6 protein and serum from patients with high CST6 significantly inhibited the activity of the osteoclast-specific protease cathepsin K and blocked osteoclast differentiation and function. Recombinant CST6 inhibited bone destruction in ex vivo and in vivo myeloma models [37]. Thus, the secretome of AML-MSCs, in contrast to normal MSCs, is unlikely to inhibit osteoclast differentiation.

A third protein is LOXL4 (Lysyl oxidase homolog 4), which may modulate the formation of a collagenous extracellular matrix. LOXL4 is an amine oxidase that is primarily involved in extracellular matrix remodeling. In vitro exposure of macrophages to LOXL4 invoked an immunosuppressive phenotype and activated programmed death ligand 1 (PD-L1) expression, which further suppressed the function of CD8^+^ T cells [38]. LOXL4 knockdown enhances tumor growth and lung metastasis through collagen-dependent extracellular matrix changes in triple-negative breast cancer [39]. Therefore, the not revealed secretion of LOXL4 in AML-MSCs could indirectly support the AML blast cells. The role of the remaining proteins not revealed in AML-MSC secretome—F10 and A1BG—is not obvious. It should be noted that all proteins mentioned above fail to restore their secretion level when remission is achieved. In particular, no secretion in R-MSCs is observed for NCAM1, secretion of CST6 is partially restored in 2 patients of 13, secretion of A1BG and LOXL4 restored in 1 patient of 13, while F10 secretion is restored in 5 patients of 13. The level of secretion of all proteins that appeared in remission remains very low. So AML-MSCs do not secrete several proteins important for maintaining normal functions.

Among the proteins secreted at lower levels in AML-MSCs versus D-MSCs are GAS6, AXL, COL6A1, TGFB, PDGFA, PDGFRB, VCAM1, and CFH.

GAS6 (Growth arrest-specific protein 6)—ligand for tyrosine-protein kinase receptors AXL, TYRO3, and MER, whose signaling is implicated in cell growth and survival, cell adhesion, and cell migration. GAS6 has important effects on hemostasis and inflammation [40]. Its deficiency affects various processes such as preventing apoptosis of endothelial cells during acidification, cytokine signaling, hepatic regeneration, gonadotropin-releasing hormone neuron survival and migration, platelet activation, or regulation of thrombotic responses. Decreased secretion of GAS6 and AXL by AML-MSCs was observed in some studies and resulted in a reduced ability of MSCs to proliferate [1].

COL6A1 (Collagen VI) is a major player in extracellular matrix biology since its deficiency alters extracellular matrix structure and biomechanical properties and leads to increased apoptosis and oxidative stress, decreased autophagy, and impaired muscle regeneration [41].

TGFB (Transforming growth factor beta) is a pleiotropic factor involved in many processes in the body associated with hematopoietic stem cells and the development of various diseases. The autocrine and paracrine effects of TGF-beta on tumor cells and the tumor microenvironment exert both positive and negative influences on cancer development [42].

PDGFA (Platelet-derived growth factor alfa) The classical PDGF polypeptide chains, PDGF-A and PDGF-B, are well-studied and known to regulate several physiological and pathophysiological processes, primarily acting on cells of mesenchymal or neuroectodermal origin [43]. It is secreted by melanoma cells and maintains the growth of malignant cells. Its deficiency may have a dual effect on MSCs and AML blasts [43].

PDGFRB (Platelet-derived growth factor receptor, beta) PDGFR-beta signaling is important for blood vessel formation and early hematopoiesis and has been implicated in a range of pathologies. Paracrine PDGF signaling triggers stromal recruitment in epithelial cancers and may be involved in epithelial–mesenchymal transition, affecting tumor growth, angiogenesis, invasion, and metastasis [44], while autocrine activation of PDGF signaling pathways is involved in gliomas, sarcomas, and leukemias.

VCAM1 (Vascular cell adhesion molecule 1) was originally identified as a cell adhesion molecule that helps regulate inflammation-associated vascular adhesion and the transendothelial migration of leukocytes, such as macrophages and T cells. Recent evidence suggests that VCAM-1 is closely associated with the progression of various immunological disorders, including rheumatoid arthritis, asthma, transplant rejection, and cancer [45].

CFH (Complement factor H) is a central regulator of early alternative pathway activation by acting as a cofactor for factor I in the cleavage of C3b into iC3b [46]. Complement deficiencies within the mannose-binding lectin pathway generally lead to increased bacterial infections, and deficiencies within the alternative pathway usually lead to an increased frequency of Neisseria infections. However, factor H deficiency can lead to membranoproliferative glomerulonephritis and hemolytic uremic syndrome [47].

A comparison of secretomes of AML-MSC and D-MSCs identifies 137 proteins secreted at significantly higher levels by AML patients’ MSCs. Proteins secreted at higher levels by AML-MSCs may indicate the possible activation of these cells. Interestingly, MSCs grown in culture for at least two passages exhibit the properties of activated cells, with this activation probably already occurring in the patient’s bone marrow. Among the upregulated proteins are those that are important for maintaining hematopoiesis and the immune response, forming relationships—HLA-A, MIF, COL8A1, POSTN, IGFB5, CD44, LGALS3 и ANXA1, CXCL12.

Three proteins, namely HLA-A (HLA class I histocompatibility antigen) and B2M (beta-2-microglobulin), are involved in the presentation of foreign antigens. It is known that in patients with non-Hodgkin’s lymphomas, plasma levels of b2 microglobulin and secreted HLA-ABC are increased [48]. Elevated levels of secreted HLA-I are found in the blood serum of patients with rheumatoid arthritis, systemic lupus erythematosus, and some viral infections [49]. It has been shown that with some solid tumors, the level of HLA-I in the blood serum of patients also increases [50]. There is evidence indicating that chemotherapy and targeted therapies are effective at enhancing HLA class I component expression and function in cancer cells [51]. It is assumed that increased secretion of HLA-I is associated with increased production of cytokines and activation of AML-MSCs in comparison with D-MSCs.

However, another protein—MIF (macrophage migration inhibitory factor)—is a pro-inflammatory cytokine involved in the innate immune response to bacterial pathogens. The expression of MIF at sites of inflammation suggests its role as a mediator in regulating the function of macrophages in host defense. MIF upregulation forms a pro-tumor microenvironment in response to hypoxia-induced factors and promotes pro-inflammatory cytokine production [52]. MIF activates CD44, which is elevated in AML-MSCs secretome. CD44 is an adhesion molecule that mediates the activation of the Src proto-oncogene protein family. MIF-activated CD44 is expressed in cells with dynamic proliferation, such as tumor cells [53]. The increased secretion of these two proteins indicates the maintenance of malignant cells by AML-MSCs.

POSTN (periostin) may regulate multiple biological behaviors of tumor cells [54]. Periostin is a member of matricellular proteins that regulate a variety of biological processes in normal and pathological situations. Many members of this family, such as periostin, osteopontin (SPP1), or the CNN (Cyr61, CCN2, CCN3) family of proteins, have been shown to regulate key aspects of tumor biology, including proliferation, invasion, matrix remodeling, and dissemination to pre-metastatic niches in distant organs [55].

IGFBP5 (insulin-like growth factor binding protein 5) is a secreted protein involved in insulin-like growth factor signaling, a critical pathway in growth and development. IGFBP5 modulates matrix signaling and can increase fibrosis and adipogenesis [56]. Elevation of IGFBP5 secretion plays a role in cancer progression and could be associated with poor prognosis.

LGALS3 (Galectin 3) is a member of the galectin family. It is predominantly located in the cytoplasm; however, it shuttles into the nucleus and is also a secreted protein. It serves important functions in numerous biological processes, including cell growth, apoptosis, pre-mRNA splicing, differentiation, transformation, angiogenesis, inflammation, fibrosis, and host defense [57]. Galectin-1, galectin-3, and galectin-9 secreted by MSCs prevent proliferation and induce apoptosis of activated Th1- and Th17-lymphocytes and CD8 + T-lymphocytes, promote the survival of naive T-lymphocytes, generation of tolerogenic dendritic cells and activation of Tregs [58]. Increased secretion of galectins also indicates changes in MSCs associated with the presence of blast cells in the bone marrow in AML patients.

CXCL12—stromal-derived factor 1 (SDF1) is a chemokine belonging to a family of small cytokines with chemotactic activity and is responsible for homing of hematopoietic cells and responsive to pro-inflammatory stimuli. Elevated serum levels of CXCL12 were found to be associated with increased osteoclast formation and bone loss in myeloma patients, and that targeted disruption of the CXCL12/CXCR4 axis inhibited osteolysis in a murine model of myeloma-associated bone loss [59]. In addition to bone marrow stromal cells, circulating plasma cells in multiple myeloma also produce CXCL12 and the high levels of CXCL12 in this disease are involved in various pathological processes [60]. Enhanced secretion of CXCL12 by AML-MSCs and R-MSCs (in the latter case at lower levels and not statistically significant) versus D-MSCs indicates a substantial but not complete recovery of the secretion of this chemokine. The interaction of CXCL12 and its receptors subsequently induces downstream signaling pathways with broad effects on chemotaxis, cell proliferation, migration, and gene expression. Accumulating evidence suggests that the CXCL12/CXCR4/CXCR7 axis plays a pivotal role in tumor development, survival, angiogenesis, metastasis, and tumor microenvironment [61].

In the remission of the disease, there is little recovery of MSCs secretome. The 220 proteins are secreted at lower levels compared to D-MSCs in both AML-MSCs and R-MSCs, while the secretion of 104 proteins is uniquely reduced in AML-MSCs and 88 proteins—in R-MSCs. The 86 proteins that were upregulated in AML-MSCs compared to D-MSCs remain elevated in remission, with R-MSCs over-secreting 37 more unique proteins compared to D-MSCs. These data indicate strong changes in the secretome associated not only with the primary adaptation of the stromal microenvironment to tumor cells but also with the damaging effect of chemotherapy. The differences in protein secretion between AML-MSCs and R-MSCs are significantly smaller as compared to their differences to D-MSCs.

It should also be noted that in a number of instances, the secretion of the most important proteins associated with the processes of protein degradation (ubiquitin proteasome system), autophagy, ferroptosis, the translation process, the organization of the extracellular matrix, and others does not change unidirectionally, and many proteins involved in these processes change the level of secretion.

The above findings indicate that changes in MSC secretome induced by AML are largely irreversible. Of note, the aging of normal MSCs ex vivo is also considered to be irreversible and results in senescent phenotype. According to some studies, AML-MSCs, both in culture and in vivo, are reminiscent of normal senescent MSCs [62,63]. We may hypothesize that extensive proliferation inherent to AML cells induces rapid wear-out and exhaustion of AML niche MSCs that support a very intense leukemia cell metabolism. If so, this is likely to result in accelerated aging of AML-MSCs in bone marrow, a process that should be additionally enhanced by chemotherapy treatments.

A significant number of ribosomal proteins and translation initiation factors are reduced or absent in the secretome of AML-MSCs compared to donor MSCs. Whether this finding signifies the reduced translation efficiency in AML-MSCs remains to be clarified. It might be highly indicative, though, that a recent study demonstrated the loss of regulation of protein synthesis and reduction of ribosomal protein levels in senescent MSCs [64].

Changes in the secretome of MSCs in AML patients may be related to the fact that leukemic stem cells survive in the bone marrow of patients after chemotherapy and bone marrow transplantation and may begin to proliferate and cause a relapse of the disease. It has been suggested that MSCs have the ability to form a cancer stem cell niche in which tumor cells can preserve the potential to proliferate and sustain the malignant process [65]. Our data, although not directly pinpointing specific changes in the AML MSC secretome to the disease recurrence, open up opportunities for further studies of the leukemic bone marrow microenvironment that may be associated with relapse.

Finally, it would be important to reiterate that, according to numerous publications, AML-educated bone marrow niche subverts normal hematopoiesis and cooperates with leukemia cells. Combined with the major finding of this study demonstrates that changes in MSC secretome induced by AML are largely irreversible; this may indicate that even in remission, MSCs retain their “subverted” phenotype and remain the “enemy within” the patient’s organism. This may, in turn, open up new therapeutic strategies aimed specifically at this “enemy within”. It should be noted that traditional therapies targeting leukemic cells have failed to improve long-term survival rates, and the bone marrow niche is thus becoming a promising source of potential therapeutic targets, particularly for relapsed and refractory AML [66]. The data of this study on differences between D-MSC, AML-MSC, and R-MSC secretomes may provide important clues to developing new therapeutic strategies. These therapies may include pro-apoptotic agents, microenvironment targeting molecules, cell cycle checkpoint inhibitors, and epigenetic regulators [67]. In addition, one could envisage approaches directed towards reverse in vivo education of subverted MSCs using exosomes and microvesicles derived from normal MSCs. An alternative strategy may involve attempts to replace AML-subverted MSCs in the niche with their normal counterparts.

## 4. Materials and Methods

### 4.1. Patients

The study included MSCs obtained from the bone marrow of 13 AML patients: (3 male, 10 female, median age 38) and 21 donors (10 male, 11 female, median age 35) that were used as control (Table 5 and Appendix A). All donors and patients signed informed consent.

### 4.2. MSCs

Donor MSCs for each patient were prepared from the bone marrow of a hematopoietic stem cell donor collected at the time of collection, as described [68]. For research purposes, MSCs from the same donors, after signing their informed consent, were further expanded as described [69].

The MSCs were derived from 2 to 8 mL of donor bone marrow. For the separation of mononuclear cells, the bone marrow was mixed with an equal volume of alpha-MEM (ICN) containing 0.2% methylcellulose (1500 cP, Sigma-Aldrich, St. Louis, MO, USA). After 40 min, most of the erythrocytes and granulocytes precipitated, while the mononuclear cells remained in the supernatant. The supernatant fraction was aspirated and centrifuged for 10 min at 450 g.

The cells from the sediment were resuspended in a standard culture medium that was composed of alpha-MEM supplemented with 10% fetal bovine serum (HyClone, Logan, UT, USA), 2 mM L-glutamine (ICN, Costa Mesa, CA, USA), 100 U/mL penicillin (Synthesis, Russia), and 50 μg/mL streptomycin (BioPharmGarant, Vladimir, Russia). The cells (3 × 10^6^) were cultured in T25 culture flasks (Corning-Costar, Corning, NY, USA). After reaching confluency, the cells were washed with 0.02% EDTA (ICN, USA) in a physiological solution (Sigma-Aldrich, USA) and then detached by 0.25% trypsin (ICN, USA) treatment (Passage 0). For expansion, the cells were seeded at 4 × 10^3^ cells per cm^2^ of the flask growth area. The cultures were maintained under hypoxia conditions at 37 °C in 5% O_2_ and 5% CO_2_. The number of harvested cells was counted directly; cell viability was checked by trypan blue dye exclusion staining.

### 4.3. Preparation of MSC-Conditioned Medium

MSCs at passages 2–3 were seeded at 4 × 10^3^ cells per cm^2^ into T175 flasks (Costar, USA). After attaining confluence (3–4 days), the flasks were washed five times with phosphate buffer without Ca^2+^/Mg^2+^ (Invitrogen, Waltham, MA, USA) and then cultured for 24 h in RPMI 1640 medium without serum and phenol red (HyClone, USA). The conditioned medium was spun off at 400 g and frozen at −70 °C.

### 4.4. Proteomic Analysis of Secretomes

For proteomic analysis, secretome samples were thawed and passed through a 25 mm Syringe Filter 0.20 µm (GVS, Findlay, OH, USA), followed by the addition of Protease Inhibitor Mix and Acetonitrile (ACN) to a final concentration of 10%. The samples were then placed in an Amicon Ultra-15 centrifuge concentrator (Millipore, Burlington, MA, USA) with a nominal molecular weight cutoff of 3 kDa and centrifuged at 5000× *g* at 4 °C for 30 min. A total of 10 mL of 50 mM Tris-HCl, pH 7.4, 150 mM NaCl, 3 mM MgCl_2_ and 10% ACN was added to the concentrated samples, mixed, and centrifuged at 5000× *g* at 4 °C for 30 min. The eluates were discarded, and this step was repeated twice. A total of 1.5 mL of the concentrated sample was removed from the filter and transferred to a new tube, and TCEP (tris(2-carboxyethyl)phosphine) and CAA (Chloroacetamide) were added to the final concentrations of 5 and 30 mM, respectively. Cysteine reduction and alkylation were achieved by incubation of the sample at 80 °C for 10 min. Proteins were precipitated by the addition of 9x volume of cold acetone and incubation at −20 °C overnight. The protein pellet was washed twice with cold acetone, followed by resuspension in 50 µL of the 100 mM Tris-HCl buffer (pH 8.5). Protein concentration was determined by BCA Assay Kit (Sigma-Aldrich, St. Louis, MO, USA). Trypsin (Trypsin Gold, Mass Spectrometry Grade, Promega, Madison, WI, USA) was added to protein samples at a ratio of 1/100 *w*/*w* and incubated for 2 h at 37 °C. Then the second trypsin portion 1/100 *w*/*w* was added, and the sample was incubated at 37 °C overnight. Proteolysis was stopped by adding TFA to 1%. Peptides were dried in a SpeedVac (Labconco, Kansas City, MO, USA) and resuspended in 20 µL of 3% ACN, 0.1% TFA in MilliQ water. The peptide concentration was determined by BCA Assay Kit (Sigma-Aldrich, St. Louis, MO, USA).

### 4.5. LC-MS/MS Analysis

Proteomic analysis was performed on an Orbitrap Q Exactive HF-X (Thermo Fisher Scientific, Waltham, MA, USA) mass spectrometer equipped with a nano-electrospray (nano-ESI) source and a high-pressure nanoflow chromatograph UPLC Ultimate 3000 (Thermo Fisher) equipped with a lab-packed reverse-phase (Kinetex C18, 2.4 μm) column (100 μm × 500 mm). The temperature of the column was thermostatically controlled at 60 °C. Samples were loaded in buffer A (0.1% Formic acid) and eluted with a linear (180 min) gradient of 3 to 55% buffer B (0.1% Formic acid, 80% Acetonitrile) at a flow rate of 220 nL/min. Mass spectrometric data were stored during automatic switching between MS1 scans and up to 16 MS/MS scans (topN method). The target value for MS1 scanning was set to 3 × 10^6^ in the range 390–1400 *m*/*z* with a maximum ion injection time of 45 ms and a resolution of 60,000. The precursor ions were isolated at a window width of 1.4 *m*/*z*. Precursor ions were fragmented by high-energy dissociation in a C-trap with a normalized collision energy of 30 eV. MS/MS scans were saved with a resolution of 15,000 at 400 *m*/*z* and a value of 2 × 10^5^ for target ions with a maximum ion injection time of 50 ms.

### 4.6. Protein Identification and Bioinformatics Analysis

Raw LC-MS/MS data from Q Exactive HF mass-spectrometer were converted to .mgf peak lists with MSConvert (ProteoWizard Software Foundation, Palo Alto, CA, USA). For this procedure, we used the following parameters: “--mgf --filter peakPicking true [1,2]”. For thorough protein identification, the generated peak lists were searched with MASCOT (version 2.5.1, Matrix Science Ltd., London, UK) and X! Tandem (ALANINE, 2017.02.01, 2017.02.01, The Global Proteome Machine Organization) search engines against UniProt human protein knowledgebase with the concatenated reverse decoy dataset. The precursor and fragment mass tolerance were set at 20 ppm and 0.04 Da, respectively. Database-searching parameters included the following: tryptic digestion with one [70] possible missed cleavage, static modification for carbamidomethyl (C), and dynamic/flexible modifications for oxidation (M). For X! Tandem, we also selected parameters that allowed a quick check for protein N-terminal residue acetylation, peptide N-terminal glutamine ammonia loss, or peptide N-terminal glutamic acid water loss. Result files were submitted to Scaffold 5 software (version 5.1.0) for validation and meta-analysis. We used the local false discovery rate scoring algorithm with standard experiment-wide protein grouping. For the evaluation of peptide and protein hits, a false discovery rate of 5% was selected for both. False positive identifications were based on reverse database analysis. We also set protein annotation preferences in Scaffold to highlight Swiss–Prot accessions among others in protein groups.

The interactions between identified differentially secreted proteins were analyzed using the STRING-db online service.

### 4.7. Statistics

Statistical analysis was performed using GraphPad Prism version 8-1 (GraphPad Software Inc., San Diego, CA, USA). Due to the non-normal distribution of the data, the Mann–Whitney U test was used for comparison. Differences were considered significant at *p* < 0.05.

PCA analysis was performed on scaled and centered protein counts within the R environment.

## 5. Conclusions

AML causes crucial changes in the MSC secretome. Some secretome modifications induced by the tumor, although not all of them, are reversed upon achieving remission, while chemotherapy leads to new alterations. Thus, the functions of bone marrow MSCs in remission remain impaired despite the absence of tumor cells and the formation of benign hematopoietic cells.

## Figures and Tables

**Figure 1 ijms-24-08953-f001:**
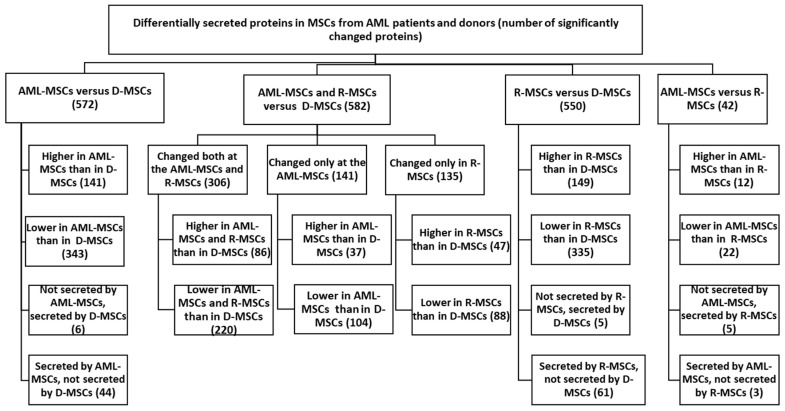
Proteins secreted at significantly different levels in MSCs from AML patients and donors. Differential expression was calculated on normalized log_2_ ratios. Differences were considered significant at *p* < 0.05, (FC ≥ 2).

**Figure 2 ijms-24-08953-f002:**
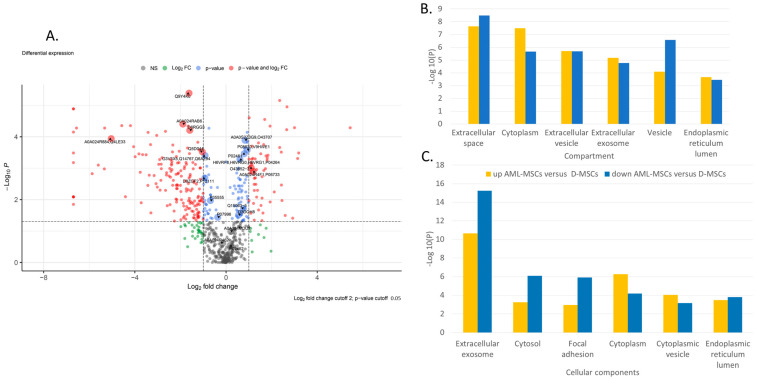
Proteome analysis of AML-MSCs versus D-MSCs (**A**) Volkano plot, (**B**) GO-term enrichment analysis for the compartment, and (**C**) GO-term enrichment analysis for and cellular component. Bar charts represent the most significant top 20 terms of each category for each cell type sorted by mean −log_10_
*p* values.

**Figure 3 ijms-24-08953-f003:**
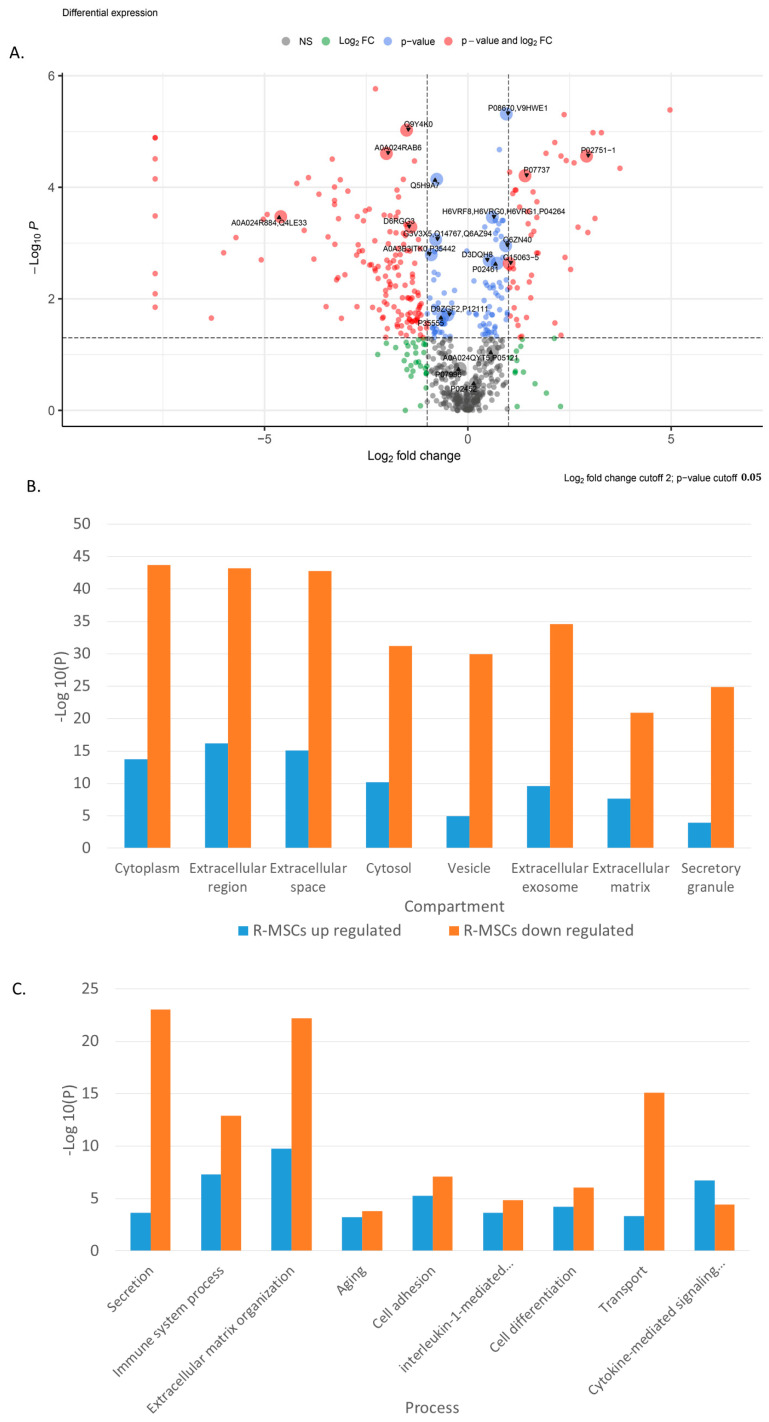
Proteome analysis of R-MSCs versus D-MSCs. (**A**) Volkano plot, (**B**) GO-term enrichment analysis for the compartment, and (**C**) GO-term enrichment analysis for processes. Bar charts represent the most significant top terms of each category for each cell group sorted by mean −log10 *p* values.

**Figure 4 ijms-24-08953-f004:**
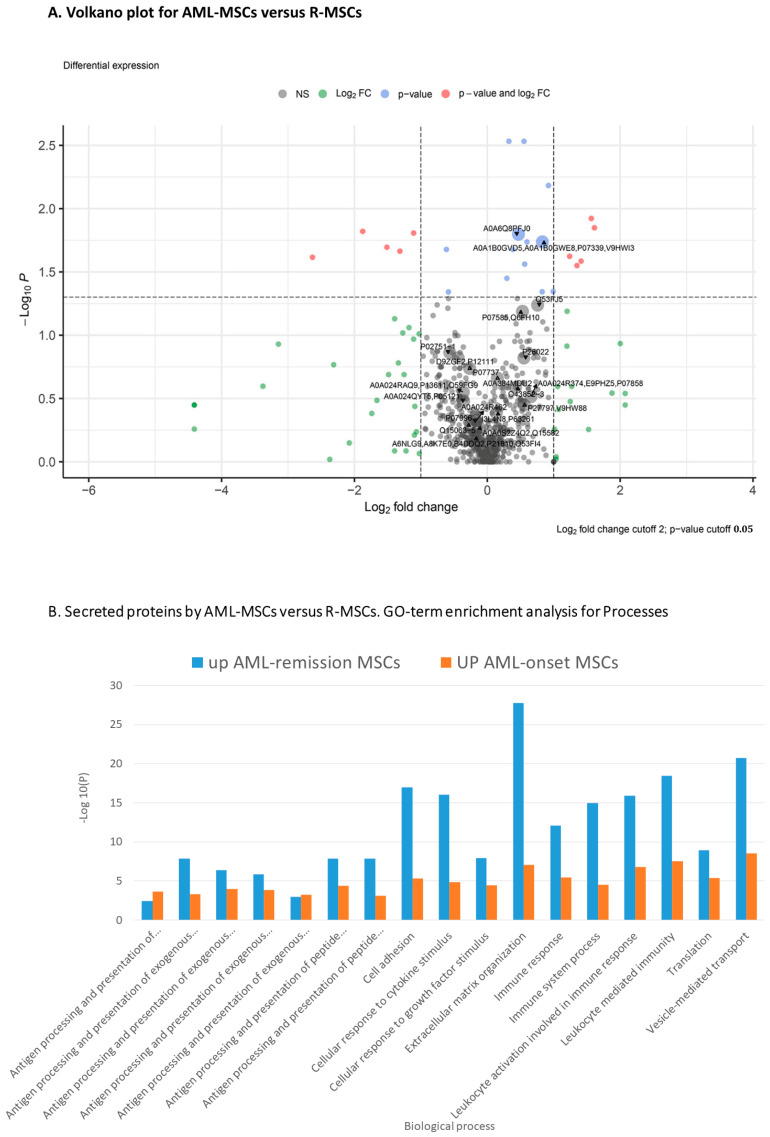
Proteome analysis of AML-MSCs versus R-MSCs (**A**) Volkano plot, (**B**) GO-term enrichment analysis for biological process.

**Table 1 ijms-24-08953-t001:** Comparison of AML-MSCs and D-MSCs secretomes. GO—Gene Ontology.

Elements	Pathway Database	Term Description
Higher in AML-MSCs versus D-MSCs	GO COMPARTMENTS	Extracellular space, Cytoplasm, Extracellular region, Extracellular vesicle, Extracellular exosome, Proteasome core complex, Intracellular, Secretory granule, Integral component of endoplasmic reticulum membrane, Vesicle, Organelle, Membrane-bounded organelle, Intracellular organelle, Endomembrane system, ficolin-1-rich granule lumen, Intracellular organelle lumen, Intracellular membrane-bounded organelle, MHC class I peptide loading complex, Tertiary granule lumen, Endoplasmic reticulum lumen, Basement membrane, Proteasome core complex, beta-subunit complex, Cytoplasmic vesicle, Cellular anatomical entity, Collagen-containing extracellular matrix, Cytosol
	GO PROCESS	Antigen processing and presentation of exogenous peptide antigen, Antigen processing and presentation of exogenous peptide antigen via MHC class I, tap-dependent, Cytokine-mediated signaling pathway, Regulated exocytosis, Proteasomal ubiquitin-independent protein catabolic process, Cellular response to cytokine stimulus, Vesicle-mediated transport, Immune system process, Secretion, interleukin-1-mediated signaling pathway, Cellular process
	GO COMPONENT	Extracellular region, Extracellular exosome, Extracellular space, Vesicle, Cytoplasm, Membrane-bounded organelle, Proteasome core complex, Secretory granule, Cytoplasmic vesicle, MHC class I peptide loading complex, ficolin-1-rich granule lumen, Tertiary granule lumen, Proteasome core complex, beta-subunit complex, Integral component of endoplasmic reticulum membrane, Endoplasmic reticulum lumen, Intracellular organelle lumen, Secretory granule lumen, Cytosol, Basement membrane, Endomembrane system, MHC protein complex, Collagen-containing extracellular matrix, Focal adhesion, Transport vesicle
Lower in AML-MSCs versus D-MSCs	GO COMPARTMENTS	Extracellular space, Intermediate filament, Supramolecular fiber, Cytoskeleton, Vesicle, Polymeric cytoskeletal fiber, Keratin filament, Cytosol, Cytoplasm, Extracellular vesicle, Focal adhesion, Anchoring junction, Cornified envelope, Extracellular exosome, Intracellular non-membrane-bounded organelle, Pseudopodium, Blood microparticle, Intracellular organelle, Endoplasmic reticulum lumen, Cell surface
	GO PROCESS	Cornification, Epithelial cell differentiation, Glycolytic process, Multicellular organism development, Peptide cross-linking, Multicellular organismal process, glyceraldehyde-3-phosphate biosynthetic process, Epithelium development, Supramolecular fiber organization, NAD metabolic process, interleukin-12-mediated signaling pathway, Monosaccharide biosynthetic process, Animal organ development, Cytoskeleton organization, Protein folding in the endoplasmic reticulum, Protein tetramerization, System development, Cell differentiation, Protein heterotetramerization, Oxidation-reduction process
	GO COMPONENT	Extracellular exosome, Cytosol, Intermediate filament, Focal adhesion, Supramolecular fiber, Melanosome, Anchoring junction, Cytoskeleton, Cornified envelope, Polymeric cytoskeletal fiber, Cytoplasm, Cell surface, Endoplasmic reticulum chaperone complex, Keratin filament, Pseudopodium, Endoplasmic reticulum lumen, Blood microparticle, Nucleus, Cytoplasmic vesicle
	GO FUNCTION	Structural molecule activity, Structural constituent of the cytoskeleton, Protein binding, Structural constituent of skin epidermis, Cell adhesion molecule binding, Cytoskeletal protein binding, Intramolecular oxidoreductase activity, Actin binding, Structural constituent of postsynapse, Peptide disulfide oxidoreductase activity, Protein disulfide isomerase activity, Identical protein binding, Cadherin binding, Protein dimerization activity

**Table 2 ijms-24-08953-t002:** Comparison of R-MSCs and D-MSCs secretomes. GO—Gene Ontology.

Elements	Pathway Database	Term Description
Higher in R-MSCs versus D-MSC	Biological process	Negative regulation of intrinsic apoptotic signaling pathway in response to DNA damage by p53 class mediator, Proteasomal ubiquitin-independent protein catabolic process, Endothelial cell development, Regulation of transcription from RNA polymerase II promoter in response to hypoxia, interleukin-1-mediated signaling pathway, Antigen processing and presentation of peptide antigen via MHC class I, Cellular response to interferon-gamma, Response to interleukin-1, Cellular response to hypoxia, T cell receptor signaling pathway, Extracellular matrix organization, Regulation of response to DNA damage stimulus, Positive regulation of growth, Cytokine-mediated signaling pathway, Immune response, Cellular response to tumor necrosis factor, Post-translational protein modification, Leukocyte mediated immunity, Response to cytokine, Cell adhesion, Secretion, Vesicle-mediated transport, Regulation of apoptotic process, Cell differentiation
Lower in R-MSCs versus D-MSCs	Biological process	Aging, Angiogenesis, Antigen processing and presentation, Blood coagulation, Blood vessel development, Bone development, Cell activation, Cell adhesion, Cell morphogenesis involved in differentiation, Chemotaxis, Chondrocyte development, Complement activation, Exocytosis, Extracellular matrix organization, Immune response, Innate immune response, Leukocyte activation, Myeloid leukocyte mediated immunity, Ossification, Plasminogen activation, Platelet-derived growth factor receptor signaling pathway, Posttranscriptional regulation of gene expression, Regeneration, Regulation of cell death, Regulation of cell differentiation, Regulation of cell growth, Signal transduction, Skeletal system development, Tissue homeostasis, Transforming growth factor beta receptor signaling pathway, Translation, Transport, Vesicle-mediated transport

**Table 3 ijms-24-08953-t003:** Comparison of AML-MSCs and R-MSCs secretomes. GO—Gene Ontology.

Elements	Pathway Database	Term Description
Higher in R-MSCs versus AML-MSCs	GO Cellular component	Endomembrane system, Endoplasmic reticulum, Endoplasmic reticulum lumen, Extracellular exosome, Extracellular region, Extracellular space, Intracellular organelle lumen, Lysosome, Melanosome, Vesicle
Lower in R-MSCs versus AML-MSCs	Cellular component	Collagen-containing extracellular matrix, Extracellular matrix, Extracellular exosome, Extracellular space, Extracellular region

**Table 4 ijms-24-08953-t004:** Secretion differences between AML-MSCs and R-MSCs versus D-MSCs. GO—Gene Ontology.

Elements	Pathway Database	Term Description
Equally expressed compared to D-MSCs		
Higher in R-MSCs and AML-MSCs versus D-MSCs	GO Biological process	Extracellular matrix assembly, Platelet degranulation, Chondrocyte differentiation, interleukin-12-mediated signaling pathway, Platelet-derived growth factor receptor signaling pathway, Osteoclast differentiation, Regulation of epithelial to mesenchymal transition, Bone development, Cellular response to transforming growth factor beta stimulus, Cell adhesion, Angiogenesis, Blood coagulation, Mesenchymal cell differentiation, Secretion, Response to growth factor, Negative regulation of canonical WNT signaling pathway, Cell activation involved in immune response, Aging, Vesicle-mediated transport, Posttranscriptional regulation of gene expression, Regulation of translation, Immune system process, Regulation of cell death, Cellular protein modification process, Signaling
Lower in R-MSCs and AML-MSCs versus D-MSCs	Biological process	Response to interleukin-1, Extracellular matrix organization, Leukocyte migration, Response to hypoxia, Angiogenesis, Cytokine-mediated signaling pathway, Blood vessel morphogenesis, Cellular response to cytokine stimulus, Leukocyte mediated immunity, Cell adhesion, Secretion, Vesicle-mediated transport, Immune system process, Cell differentiation
Differently expressed compared to D-MSCs		
Secreted by R-MSCs Lower than D-MSCs	Biological process	Regulation of complement activation, Cell-matrix adhesion, Leukocyte mediated immunity, Extracellular matrix organization, Angiogenesis, Secretion, Immune response, Vesicle-mediated transport, Cell adhesion
Secreted by R-MSCs Higher than D-MSCs	Cellular component	Extracellular exosome, Extracellular space, Extracellular region, Vesicle
Secreted by AML-MSCs Lower than D-MSCs	Biological process	interleukin-12-mediated signaling pathway, Extracellular matrix organization, Osteoblast differentiation, Ossification, Transmembrane receptor protein tyrosine kinase signaling pathway, Cell adhesion, Response to cytokine, Response to growth factor, Secretion, Vesicle-mediated transport, Immune system process
Secreted by AML-MSCs Higher than D-MSCs	Cellular component	Cytosolic ribosome, Secretory granule lumen, Collagen-containing extracellular matrix, Extracellular matrix, Secretory granule, Extracellular exosome, Extracellular space, Extracellular region, Vesicle

**Table 5 ijms-24-08953-t005:** Characteristics of patients and donors.

	Acute Myeloid Leukemia		Donors
	Onset of the Disease	Remission	
Age, years (median)	26–64 (38)	26–64 (38)	22–59 (36)
Gender male/female	3/10	3/10	10/11
Cumulative MSCs production for 3 passages, × 10^6^ (M ± ME)	12.5 ± 3.2	15.3 ± 2.2	8.7 ± 1.7
MSCs-Time to P0, days (M ± ME)	15.8 ± 1.2	14.1 ± 0.9	12.3 ± 0.4
MSCs-Time to P3, days (M ± ME)	35.2 ± 1.9	29.2 ± 1.5	24.4 ± 0.8

## Data Availability

The data presented in this study are available from the corresponding author upon reasonable request.

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
