# Peer review of "Acute Myeloid Leukemia Causes Serious and Partially Irreversible Changes in Secretomes of Bone Marrow Multipotent Mesenchymal Stromal Cells"

_ijms, 2023, doi:10.3390/ijms24108953_

Round 1
Reviewer 1 Report
Did the authors provide any functional characterization of MSCs besides proliferation?
How did the serum deprivation affect proteomic profiles of secretomes?
Where any unique proteins detected in AML vs D, R vs D?
Author Response
We woul like to thank the reviewer.

Reviewer 2 Report
This research manuscript is well written, interesting and covers a very relevant topic. The study is well executed.
Suggestion:
Perform imaging of MSCs: Is there a morphological difference between MSCs obtained from AML patients at diagnosis, AML patients in remission, and healthy donors?
Introduction:
- Line 43: How are MSCs related to CAR cells, nestin+ cells and CD46+ cells? Which cell types are nestin-positive and CD46-positive in bone marrow? Please elaborate
- Line 56, when discussing receptor-adhesion molecule interactions in bone marrow and line 70, when discussing leukemic stem cells in HSC niches in bone marrow: the clinical impact of these statements can be amplified by mentioning that disruption of the receptor-adhesion molecule interactions between HSC niches and leukemic stem cells might be a therapeutic target as discussed in Hira et al, Novel therapeutic strategies to target leukemic cells that hijack compartmentalized continuous hematopoietic stem cell niches. Biochim Biophys Acta Rev Cancer. 2017 Aug;1868(1):183-198. doi: 10.1016/j.bbcan.2017.03.010 and early results of this strategy are summarized in Maganti et al, Plerixafor in combination with chemotherapy and/or hematopoietic cell transplantation to treat acute leukemia: A systematic review and metanalysis of preclinical and clinical studies. Leuk Res 2020 Oct;97:106442. doi: 10.1016/j.leukres.2020.106442
- Line 74: Please clarify this sentence into (for example): MSCs isolated from AML patients generate various factors that stimulate AML cell homing into hematopoietic stem cell niches in bone marrow, resulting in AML cell survival.
Results:
- Please change the orientation of Fig 3 and Fig 4 so that the legend fonts can be enlarged.
Discussion:
- How can the acquired AML-MSC data be used to develop novel therapeutic stategies against AML? For example, add a short discussion or list of clinical trials that are related to MSCs or their secreted proteins in AML.
Materials & methods:
- Please add more information regarding the included AML patients. What where their disease characteristics according to ELN stage? What was the marrow blast % at the time of harvesting of MSCs at disease onset? Which what regimen where they brought in remission? Were all remissions complete? Were all cells collected in CR1, or some in CR2?
Minor editing of English language required
Author Response
We would like to thank the reviewer.
